# A Single-Copter UWB-Ranging-Based Localization System Extendable to a Swarm of Drones

Christoph Steup *,† , Jonathan Beckhaus † and Sanaz Mostaghim

Faculty of Computer Science, Otto von Guericke University Magdeburg, 39106 Magdeburg, Germany;
jonathan.beckhaus@ovgu.de (J.B.); sanaz.mostaghim@ovgu.de (S.M.)
* Correspondence: steup@ovgu.de; Tel.: +49-391-67-51021
† These authors contributed equally to this work.

**Abstract:** This paper presents a single-copter localization system as a first step towards a scalable multihop drone swarm localization system. The drone was equipped with ultrawideband (UWB) transceiver modules, which can be used for communication, as well as distance measurement. The location of the drone was detected based on fixed anchor points using a single type of UWB transceiver. Our aim is to create a swarm localization system that enables drones to switch their role between an active swarm member and an anchor node to enhance the localization of the whole swarm. To this end, this paper presents our current baseline localization system and its performance regarding single-drone localization with fixed anchors and its integration into our current modular quadcopters, which was designed to be easily extendable to a swarm localization system. The distance between each drone and the anchors was measured periodically, and a specially tailored gradient descent algorithm was used to solve the resulting nonlinear optimization problem. Additional copter and wireless-specific adaptations were performed to enhance the robustness. The system was tested with a Vicon system as a position reference and showed a high precision of 0.2 m with an update rate of <10 Hz. Additionally, the system was integrated into the FINken copters of the SwarmLab and evaluated in multiple outdoor scenarios. These scenarios showed the generic usability of the approach, even though no accurate precision measurement was possible.

**Keywords:** localization; communication; drone swarm



## 1. Introduction

Drones are gaining increasing importance in industrial and research applications. Applications range from inspection tasks for bridges [1] and wind turbines [2] to natural disaster management [3]. For these tasks, reliability, autonomy, and stability are the most relevant properties in the current research [4–6]. Currently, there is a trend towards the combination of multiple drones into swarms to enhance the performance, stability, and mission times [7–9]. Independent of the number of drones, the quality of position information is highly relevant, because it enables reliable and robust movement of the swarm. However, such reliable position information is hard to achieve. Global Positioning System (GPS) localization is not always available or precise. Indoor scenarios especially have no access to GPS, but even in outdoor environments, GPS may be unreliable or unavailable. Consequently, a general-purpose drone swarm needs a localization mechanism that is compatible with indoor and outdoor scenarios, without specific deployment.

The transition from singular drones to swarms of drones creates new challenges, especially regarding the localization, because the precision of the provided position information needs to be higher to prevent crashes among swarm members. Additionally, the localization needs to be reliable and uncertainty-aware. Reliable localization allows the swarm to execute its mission, while uncertainty awareness allows the swarm behavior to react to a degradation in the localization precision. The last important factor is scalability because most localization systems require communication among the swarm members, which is

performed through a shared medium. Consequently, using the available resources of the communication system efficiently is of utmost importance for any localization system to be scalable to any swarm size.

In a swarm of drones, more options are available for the localization of individual drones. Drones without a connection to static anchors or GPS may use multihop localization to acquire the localization information of neighbors to enhance their own localization.

Our vision is to create a swarm localization system that enables drones to switch their role between active swarm member and anchor node to enhance the localization of the whole swarm. To this end, this paper present our current baseline localization system, which was designed to be easily extendable to a swarm localization system. Additionally, its performance regarding single-drone localization with fixed anchors and its integration into our current modular quadcopters is presented.

The following section (Section 2) discusses some relevant work in the area of robot and quadcopter localization, followed by the description of the hardware used (Section 3) and our algorithms in Section 4 and the evaluation experiments and results in Section 5. The paper closes with the conclusion and future work in Section 6.

## 2. Related Work

The area of localization is filled with a plethora of approaches, because the application field is also very vast. The most prominent solution for outdoor applications is GPS. Baseline GPS is a receiver-only system, which uses satellites around the globe to provide Time-of-Flight (TOF) distances between the receiver and sending satellite. The existing known orbital data of the satellites allow the reconstruction of the position using these TOF distances and an accurate time stamp. The typical precision of GPS positions is in the order of 5 m (https://www.gps.gov/systems/gps/performance/accuracy/, accessed on 25 August 2021). However, GPS is prone to distortions if the line-of-sight is blocked. Assisted GPS (AGPS) is one approach to enhance the quality of the localization information. AGPS uses additional satellites, which are deployed regionally and compensate for the atmospheric distortions of the GPS signals. Most commercially available GPS receivers already support AGPS, which provides these receivers with a maximum accuracy of 30 cm.

Further localization enhancements of GPS for high-precision applications are available through Differential GPS (D-GPS), which uses an additional local receiver to compensate for local signal distortions. Even though D-GPS at best can reach millimeter precision, the hardware is costly and requires a long time to be deployed (more than 20 min of initial convergence time [10]). An example is the DJI D-RTK2, which costs EUR ≈ 3000 (https://store.dji.com/de/product/d-rtk-2-high-precision-gnss-mobile-station, accessed on 25 August 2021). An overview of D-GPS systems and their performance can be found in [11].

Another approach more tailored to indoor scenarios is camera-based localization systems. In this area, multiple approaches exist. The most famous one is the Vicon system, which uses infrared reflective passive markers and multiple infrared cameras to track an object with high precision (typically less than 1 mm with an update rate of 100 Hz [12]). The performance of the Vicon system comes at a very high cost of typically more than EUR 50,000. Other approaches are based on single cameras and use no markers, such as that in [13]. These provide far less precision (typically 10 cm, but also with a high update rate of 50 Hz or more). These systems are less expensive because fewer cameras and no markers are needed. A good overview of vision-based localization systems is given in [14].

The third type of system uses radiofrequency (RF) or sonar waves to detect the distance between two objects. In this area, sonar waves have been used for a long time. However, currently, the trend is moving towards extensions of existing wireless communication standards. As the information, ToF, Time-Difference of Arrival (TDoA), Received Signal Strength (RSSI), or Phase Difference (PD) is used. Among these approaches, RSSI methods provide the worst precision of ≈1.2 m for Bluetooth [15] and 1 m for WiFi [16], with the benefit that they can be applied to all existing wireless hardware without any modification. ToF,

TDoA, and PD provide better distance estimations, but typically require specialized hardware, such as the DWM1000 (https://www.decawave.com/product/dwm1000-module, accessed on 25 August 2021) modules. These modules are typically used with two-way-ranging, which sends 3 to 4 packets between two nodes, which cancel out internal timing errors and increase the precision at the cost of more communication per measurement [17]. A system using this approach was presented in [18]. The proposed system showed an average precision of 0.4 m with a 20 Hz update rate using 27 packets per measurement. This disqualifies the approach for large amounts of swarm members or anchor nodes.

The last type of approach is swarm-based, which exploits the existence of multiple objects of similar type to be localized to share information and increase precision. One such approach is called Simultaneous Localization and Optimization [19], which exploits movement command information in each object together with distance measurements between the objects to enhance the positioning precision and resolve ambiguities, as well as enhance the execution of the movement goals. The most complete approach in this area is OmniSwarm [20]. This approach provides up to 0.02 m precision for a three-drone system and 0.14 m precision for a swarm of two drones. The system uses visual odometry together with Ultra-Wideband (UWB) communication and distance estimation to determine the position of the swarm members. Even though the OmniSwarm approach provides superior localization quality, it does not scale well, due to the communication necessary to exchange information. Additionally, the system uses information from many subcomponents of a drone, which make its integration into an existing drone setup very tedious and difficult.

## 3. Hardware Design and Integration

This section gives an overview of the used hardware components and their software integration.

The localization system is integrated into our swarm of quadcopters FINken [21], which are self developed and modular consisting of a basic copter frame with four motors in X-setup. The central unit of control is a LisaMX Autopilot (https://wiki.paparazziuav.org/wiki/Lisa/MX, accessed on 25 August 2021) running Paparazzi as flight software. The autopilot also provides all necessary sensors. During our tests, the drone was equipped with a TeraRanger, which is an optical height sensor with a range of 14 m and an update rate of 1 kHz (Figure 1b). The drone is additionally equipped with a GPS module (Figure 1a), a ranging module capable of UWB communication and a micro-SD card reader used for logging.

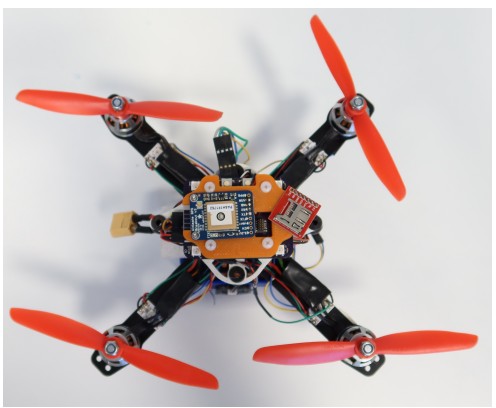
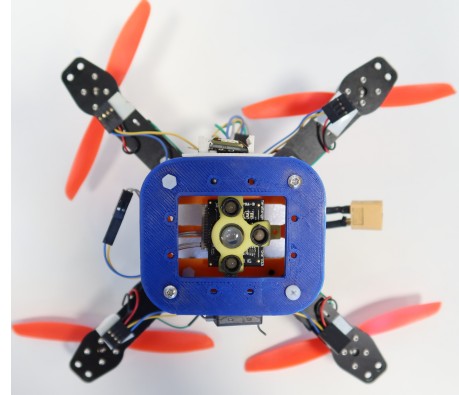

(**a**) Top view showing GPS receiver and SD card mount.

(**b**) Bottom view showing the TeraRanger distance sensor.

**Figure 1.** Images of the custom-built drones of the FINken swarm.

The communication and localization of the drones are powered by a DWM1000 UWB-compliant wireless transceiver. The DWM1000 module supports a wide frequency range from 3.5 GHz to 6.5 GHz and directly supports ToF measurement. The modules are used

to provide drones with communication capabilities between each other and to the ground station. Additionally, they provide the distance measurements for the localization system. To integrate these DWM1000 modules into our drones, a PCB was designed; see Figure 2. Besides the DWM1000 module, it contains a STM32 microcontroller, a micro-USB port and several status LEDs. In terms of connectivity, two Universal Serial Asynchronous Receiver Transmitter (USART) connections and one Inter-Integrated-Circuit ($I^2C$)-Bus, along with several input and output pins, are accessible. The modules are used as components of the copters as well as anchor nodes. This enables an easy integration of dynamic anchors created by stationary copters in later stages of the localization system. The modules do not use any specific directional antenna to enhance communications to the anchors because in the later stages of the development of the swarm localization system, the anchors will be moving.

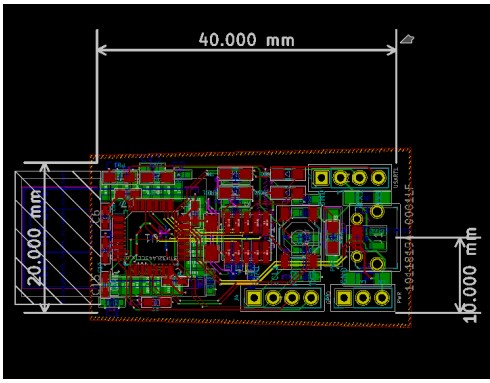

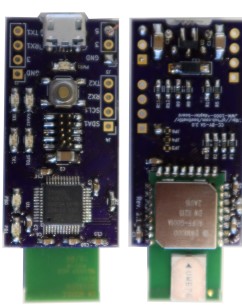

(**a**) The designed PCB of the custom interface module used for integration into the copters.

(**b**) Picture of the final manually assembled interface board.

**Figure 2.** Illustrations of the custom DWM1000 PCBs used for copters and anchors.

In addition to the distance measurements, the modules are used to transmit telemetry data of the copters to the ground station through the UWB link. The ground station is equipped with an anchor node to receive all telemetry data. To communicate with the drones during operation, the drones receive and send Paparazzi messages to the microcontroller on the DWM1000-board via USART. The microcontroller then handles the sending and receiving of data using the DWM1000 module. Furthermore, double-sided two-way ranging is implemented on the microcontroller. The modules mounted on the copters will periodically initiate two-way ranging to other nodes, while modules acting as anchor nodes will only respond to ranging requests. The microcontroller on the module then calculates a position according to our proposed localization Algorithm 1. Once the calculation is performed, a National Marine Electronics Association (NMEA)-GPS message based on that position is generated and sent to a dedicated GPS port on the autopilot.

By emulating a real GPS module, the module is easily exchangeable against a real GPS. Furthermore, existing hardware and software structures, especially the Inertial Navigation System (INS) filters of Paparazzi, can be reused. To support other researchers using Paparazzi as autopilot software, we provide the hardware description of our interface board and the software of the interface board as Open Hardware/Source through our GitHub Repository (https://github.com/ovgu-FINken/DWM1000_Copter_Integration, accessed on 25 August 2021).

---

**Algorithm 1:** Continuous Localization.

**Data:** $p^{(0)}, v_{max}, \Delta T$
**Input:** $d^{(t-1)}, \Delta t^{(t)},$
**Output:** $p^{(t)}$
**begin**

    $p^{(t)} \longleftarrow p^{(0)};$
    $m \longleftarrow 0;$
    **while** *True* **do**

        $p^{(t-1)} \longleftarrow p^{(t)};$
        $w \longleftarrow b^{\left(1 - \frac{\Delta t^{(t)}}{\Delta T}\right)};$
        $w \longleftarrow w \frac{\#w}{\sum w};$
        $p^{(t)} \longleftarrow \texttt{localization\_step}\left(p^{(t-1)}, d^{(t)}, w^{(t)}\right);$
        $v \longleftarrow \frac{\|p^{(t)} - p^{(t-1)}\|}{\Delta T};$
        **if** $v > a^m \cdot v_{max}$ **then**
            $m \longleftarrow 0;$
            $p^{(t)} \longleftarrow p^{(t-1)};$
        **else**
            $m \longleftarrow m + 1;$

---

Our localization modules differ in the generic design relating to other approaches. We combined the ranging capability of the modules with the communication capability to use the module for localization and for drone to drone and drone to ground station communication. Additionally, we did not develop a new software interface to the autopilot of the copter, but reused the existing NMEA-capable GPS interface of Paparazzi, which enables an integration of the modules to any Paparazzi-enabled copter. Finally, we did not use any specific antenna setup to not jeopardize the reception quality as soon as anchors are moving.

## 4. UWB-Ranging-Based Localization Algorithm

In the following sections, we describe the problem of the localization based on distances acquired through the UWB-Ranging modules (Section 4.1). Afterwards, we present the algorithm to compute a single localization step in Section 4.2, followed by the algorithm using the single step localization for continuous localization in Section 4.3.

### 4.1. Problem Statement

Given the observed distance values $d^{(t)} = \left(\hat{d}_0^{(t)} + e_0^{(t)}, \ldots, \hat{d}_m^{(t)} + e_m^{(t)}\right)$ between the node to be localized and each anchor node $A_i$ at time step $t$, compute the position of the node to be localized $p^{(t)}$ at time step $t$. The observed distances are composed of the correct distance $\hat{d}_i^{(t)} = \|\hat{p}^t - A_i\|$ and an unknown time-varying error $e_i^{(t)}$. Considering we have enough anchor nodes available, the resulting system of equations is over-determined. Therefore, the problem is an optimization problem as follows:

**Problem 1.** *Considering the observed distances $d^{(t)}$, we want to compute a position $p^{(t)}$, which minimizes the difference between the observed distances $d^{(t)}$ and the distance given by the estimated position and each anchor node $\|p^{(t)} - A_i\|$.*

$$\min_{p^{(t)}} \left( \sum_{i=0}^{m} \left( \|p^{(t)|} - A_i\| - d_i^{(t)} \right)^2 \right)$$

This problem is a nonlinear least squares optimization problem.

*4.2. Single Step Localization Algorithm*

To solve the optimization Problem 1 stated in Section 4.1, various approaches are possible. We decided to use a gradient-based optimization, due to its speed and the low-resource consumption. This approach is based on the work of Mantilla-Gaviria et al. [22] and Murphy and Hereman [23].

Algorithm 2 uses static ($A, \eta, z_{min}, \Delta p_{min}$ and $n$) and dynamic ($p^{(t-1)}$, $w^{(t)}$ and $d^{(t)}$) inputs. $A$ represents a $m \times 3$ matrix of the position of the $m$ anchor nodes. $\eta$ controls the speed of convergence of the gradient descent and is typically called step size. $z_{min}$ is a parameter defining the minimum height of the trajectory of the node to be tracked. This parameter is necessary to decide between multiple possible solutions in the case of planar setups of anchor nodes; see Section 5.2. $\Delta p_{min}$ and $n$ are parameters controlling the iteration of the algorithm. $\Delta p_{min}$ defines the minimum movement of the position that the algorithm needs to execute on $p^{(t)}$. If the movement is smaller, the estimated position is assumed to have converged for time step $t$ and the iteration is stopped. The iteration is stopped anyway if $n$ number of iterations are executed. $p^{(t-1)} = \left(x^{(t-1)}, y^{(t-1)}, z^{(t-1)}\right)^T$ is the estimated position of the last time step $t - 1$, and $d^{(t)}$ is the observed distance of the current time step, which is weighted by the age weight vector $w^{(t)} \in (0, 1]^m$. The output of the algorithm is the estimated position $p^{(t)} = \left(x^{(t)}, y^{(t)}, z^{(t)}\right)^T$ for time step $t$.

---

**Algorithm 2:** Single Localization Step.

**Data:** $A, \eta, z_{min}, \Delta p_{min}, n$
**Input:** $p^{(t-1)}, d^{(t)}, w^{(t)}$
**Output:** $p^{(t)}$
**begin**

    $p^{(t)} \longleftarrow p^{(t-1)}$;
    $i \longleftarrow 0$;
    **repeat**
        **for** $i$ **in** $0, \ldots, m$ **do**
            $G_i \longleftarrow p^{(t)} - A_i$;
            $R_i \longleftarrow \|G_i\|$;
            $G_i \longleftarrow \frac{G_i}{R_i}$;
        $\Delta p \longleftarrow G^{-1}\left(\left(d^{(t)} - R\right)w^{(t)}\right)$;
        $p^{(t)} \longleftarrow p^{(t)} + \eta \cdot \Delta p$;
        **if** $z^{(t)} < z_{min}$ **then**
            $z^{(t)} \longleftarrow 2z_{min} - z^{(t)}$;
        $j \longleftarrow j + 1$;
    **until** $j \geq n$ **or** $\|\Delta p\| < \Delta p_{min}$;

---

The algorithm starts by generating a gradient matrix through the computation of the vector distance between the current estimation $p^{(t)}$ and the position of each anchor node $A_i$. By computing the norm of each line $G_i$ within $G$, we obtain the estimated distance vector $R = \left(\|p^t - A_0\|, \ldots, \|p^t - A_m\|\right)^T$. We now normalize $G_i$ with $R_i$. Afterwards, the next movement $\Delta p$ of the estimated position $p^{(t)}$ is computed by pseudo-inverting $G$ and multiplying with the weighted difference of observed distances and estimated distances $\left(d^{(t)} - R\right)w^{(t)}$. If the weighting of nodes is not wanted, the weight vector $w^{(t)}$ can be replaced with a vector of ones. The use of the pseudo-inverse guarantees a correct weighting of the individual anchors to minimize the overall error. The resulting movement $\Delta p$ is scaled by the step size $\eta$ and added to the current estimated position $p^{(t)}$. If the resulting $z$-coordinate is below the minimum $z$-coordinate defined by $z_{min}$, we invert the coordinate by computing $z' = z_{min} + z_{min} - z = 2z_{min} - z$. This is repeated until either

the maximum number of iterations $n$ is reached or the current movement of the estimated position $\Delta p$ is smaller than the defined minimum movement $\Delta p_{min}$.

*4.3. Continuous Localization*

Localizing a node is a continuous process. Therefore, the Localization Step described in Algorithm 2 needs to be executed repeatedly. Algorithm 1 describes the process. This algorithm takes the current observed distances $d^{(t)}$ and the age of each distance value $\Delta t^{(t)}$ as input because we cannot expect to receive distances from all anchors at every time step. Additionally, $v_{max}$ defines the maximum expected speed of the node to be tracked, $a$ defines the base of the exponential growth of $v_{max}$ and $\Delta T$ indicates the desired update frequency of the estimated position. $b$ controls the decay of the influence of an anchor based on the age of its last received distance measurement. Higher values of $b$ increase the decay, whereas the minimal value of 1 disables it.

The algorithm starts with initializing the position of the tracked node to $p^{(0)} = (0,0,0)^T$. This initialization is arbitrary and can be adapted as necessary based on environmental circumstances. However, for our relative localization, it worked well. For each update of distance observations $d^{(t)}$, the algorithm executes `localization_step`, supplying the observed distances and the current position estimate. The weight vector regarding the age of the measurements $w$ is computed as exponentially decreasing with growing age. To prevent different sizes of update steps due to delays in communication, the weight vector is normalized to have the sum $\sum w$ equal to its length $\#w$. Afterwards, the speed of the node is estimated based on the results of the localization of the current $p^{(t)}$ and the previous time step $p^{(t-1)}$. If the estimated speed $v = \|p^{(t)} - p^{(t-1)}\|/\Delta T$ is smaller than the maximum speed $v_{max}$ multiplied by the exponential growth factor $a^m$, the position is updated. Otherwise, the old position is kept because the newly estimated position is considered an outlier. The exponential growth factor $a^m$ increases every time a position is not updated to prevent the localization from becoming stuck in local optima far away from the real trajectory due to a sequence of bad measurements.

Compared to existing systems using only ranging information, our approach is specially tailored towards drone localization. First, we assume the localization to output 3D-coordinates because drones typically move in all 3 dimensions. Additionally, we assume anchors to often be located in a plane, which poses additional challenges regarding the $z$-coordinate estimation of the algorithm. To overcome this, we use the additional $z_{min}$ parameter and the associated logic to select the correct solution. The $v_{max}$ and $a$ parameters handle the specific movement speed of drones by filtering out unreasonable values based on the possible movement speed of the tracked object. This parameter may even be delivered by the control software of the drone to further enhance the filter characteristics. To prevent the localization from becoming stuck in local optimum far away from the real trajectory, the parameter $a$ enables exponential growth of the allowed $v_{max}$ enabling resynchronization of the localization and the real trajectory.

## 5. Experimental Evaluation

To evaluate the performance of the proposed localization system, we conducted multiple experiments. We started with an experiment to evaluate the performance of the GPS receiver used in the copters to evaluate their usage as position reference, see Section 5.1. Afterward, we evaluated the system in an indoor environment equipped with a Vicon indoor localization system; see Section 5.2. Finally, we tested the system in-flight in an outdoor scenario; see Section 5.3.

*5.1. GPS Experiments*

Our first experiment was designed to test the quality of our used GPS receivers as a position reference. Additionally, we wanted to detect the level of quality the localization system needs to be beneficial to the copters in an outdoor scenario. According to our vision, our swarm of copters flies between indoor and outdoor scenarios, which places a

large quantity of the mission trajectory close to buildings. Therefore, for the GPS quality assessment, we decided to put the receiver close to the outer wall of our lab. The receiver was connected to a Raspberry Pi to collect data and write the data to an SD card. We let the receiver collect data for 7 days. The resulting distribution of positions is visible in Figure 3. Additionally, we show the distribution of the individual coordinates and the number of satellites in Figure 4a,b.

The results of the experiments show a deviation of typically ±40 m in the GPS position, even through we received data from at least 5 satellites in 99.9% of the data points. Another disadvantage is the maximum deviation of the positions, which can be as large as ±150 m. Especially noteworthy is the biased distribution of data points. The deviation is far larger for the latitude than for the longitude. This is to be expected as the receiver was placed to the north of a building, which probably reflected the incoming satellite signals and disturbed the distance measurement. This is also an explanation for the offset of the center of the measured positions against the location of the receiver.

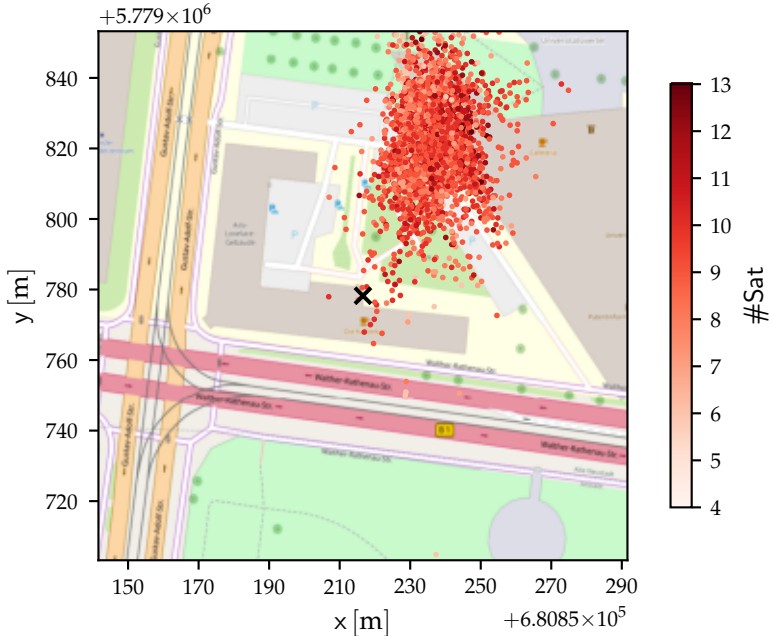

**Figure 3.** Visualization of every 100th coordinate of the static GPS receiver experiment. All coordinates are transformed into Universal Transverse Mercator (UTM) coordinates (Zone 32U) for better assessment of the distance to the center. The color indicates the number of satellites perceived by the receiver. The black **X** marks the position of the receiver.

In consequence, even a medium precision (±0.5 m) localization system used within a swarm will provide major benefits regarding swarm stability if the swarm is flying in an urban scenario. This is caused by the usage of distances between swarm members in most swarm algorithms. These distances are computed from positions by periodically calculating the difference between positions, which mathematically resembles a numerical differentiation. Stochastic variations in the position data will be amplified by the numerical differentiation, leading to large movements within the swarm, which may jeopardize swarm stability.

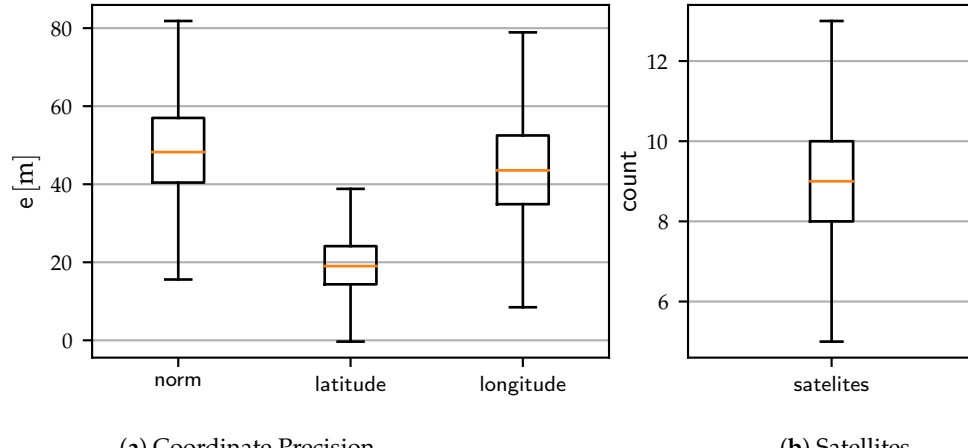

(**a**) Coordinate Precision.        (**b**) Satellites.

**Figure 4.** Box–Whisker plots showing the distribution of the longitude and latitude coordinates after transformation to UTM, as well as the distribution of satellite reception.

### 5.2. Indoor Evaluation with Precise Reference

For the precise evaluation of the localization system, we used an existing Vicon system. The Vicon camera-based tracking system delivers 100 Hz updates with a precision of 1 mm. The observable area is a cube of $3\,\text{m} \times 3\,\text{m} \times 3\,\text{m}$. We set up 8 static anchor nodes labeled 128–135 in the area, as shown in Figure 5. Due to the flat ground in the test area, all anchors were placed at 0.2 m height. This setup is typical for manually placed anchors in flat environments. This presents a problem for the localization system, as two solutions exists, which are equally likely: one above the anchors and one below the anchors. Consequently, the $z_{min}$ parameter of Algorithm 2 was set to 0.2 m to mitigate this problem for the following experiments. We executed two random trajectories by moving a single node in the area manually. The resulting trajectories are visualized in Figure 5. The node was attached to a stick of 1 m length to avoid blocking the line of sight between the node and the anchor nodes with our bodies.

Our localization system was configured with the parameters described in Table 1. The step size parameter $\eta$ was deduced as iterative through preliminary experiments. The origin of the local coordinate system was chosen as the initial position $p^{(0)}$. The minimum localization height $z_{min}$ was chosen based on the average height of the anchors. The maximum speed $v_{max}$ is defined by our copters, even though a much smaller value is defined as the maximum speed in software. The $a$ parameter was empirically deduced. $\Delta T$ was chosen based on the capabilities of our hardware modules. We derived some relevant parameter configurations to test different components of the algorithm. The default configuration contains the typical standard parameters without the use of age information. The age configuration adds the capability of weighting the individual anchors based on the age of the information. The best parameter set is the combination of all positive parameter changes in comparison to the default. The heavy configuration tests if additional computational resources may enhance the performance of the algorithm. The ground configuration checks the performance of the algorithm in case no assumption on anchor positions is made. To this end, the minimum $z$-height is defined as 0 m in the local coordinate system of the localization. The slow configuration tests the influence of the speed limitation through $v_{max}$.

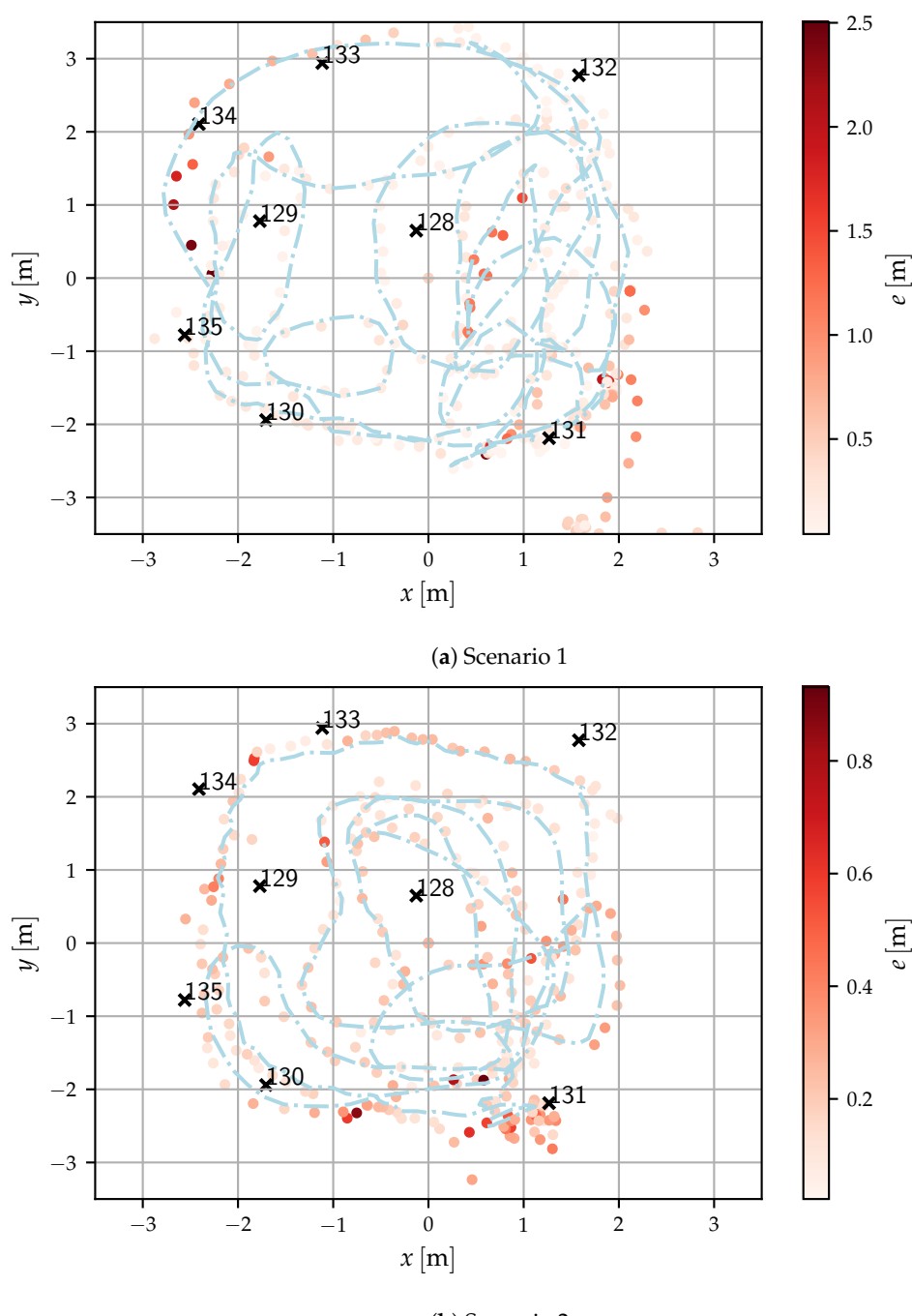

(**a**) Scenario 1

(**b**) Scenario 2

**Figure 5.** Visualizations of the difference between the Vicon reference positions (light blue dashed trajectory) and the estimated position by the localization system (red dots) in the *xy*-plane. The colors of the dots indicate the differences between the reference position and the estimated position.

**Table 1.** Localization algorithm parameter configurations. The table shows the evaluated configurations of the algorithm parameters for both algorithms. The differences of the individual configuration against the default configuration is highlighted in bold.

| Config | Algorithm 2 Parameters | | | | | Algorithm 1 Parameters | | | |
|--------|------------------------|---|---|---|---|------------------------|---|---|---|
| | $p^{(0)}$ [m] | $\eta$ | $z_{min}$ [m] | $\Delta p$ [m] | $n$ | $v_{max}$ [m/s] | $a$ | $b$ | $\Delta T$ [s] |
| default | $(0,0,0)^T$ | 0.1 | 0.2 | 0.1 | 100 | 20 | 1.01 | 1 | 0.1 |
| age | $(0,0,0)^T$ | 0.1 | 0.2 | 0.1 | 100 | 20 | 1.01 | **2** | 0.1 |
| ground | $(0,0,0)^T$ | 0.1 | **0.0** | 0.1 | 100 | 20 | 1.01 | 1 | 0.1 |
| slow | $(0,0,0)^T$ | 0.1 | 0.2 | 0.1 | 100 | **5** | **1.1** | 1 | 0.1 |
| heavy | $(0,0,0)^T$ | 0.1 | 0.2 | **0.01** | **1000** | 20 | 1.01 | 1 | 0.1 |
| best | $(0,0,0)^T$ | 0.1 | **0.0** | 0.1 | 100 | **5** | **1.1** | **2** | 0.1 |

First, we look at the difference in distance measurements between the Vicon system and our UWB-Ranging nodes. Figure 6a,b show the distance error $e$ for each trajectory for each anchor node. Interestingly, the distribution of errors is close to a zero mean with generally less than 0.1 m deviation. However, the spread of the distribution is quite high. The 90% quantile reaches $\pm 0.3$ m for both experiments. Anchor 134 is an exception with a negative 90% quantile of over $-0.5$ m. In general, the distance estimation can be assumed to provide an accuracy of $\pm 0.3$ m in most cases.

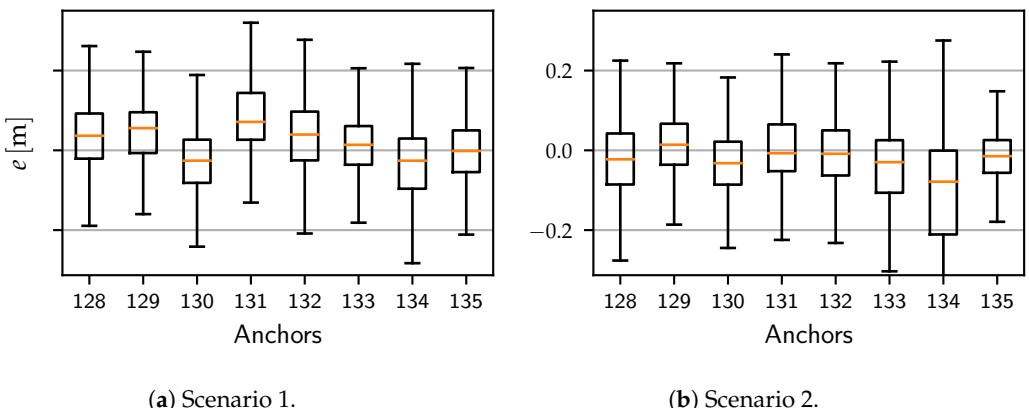

(**a**) Scenario 1.　　　　　　　　　　　　(**b**) Scenario 2.

**Figure 6.** Box–Whisker of distance errors for each anchor in both indoor evaluation scenarios.

The resulting trajectories of applying Algorithms 1 and 2 to the acquired distance values extracted from the UWB modules during the execution of trajectory 1 and 2 are shown in Figure 7. As shown in Figure 7, the error $e$ between the true positions and the estimated positions from the localization system is typically in the range of $\pm 0.25$ m. However, some outliers exist with large errors beyond $\pm 2$ m. Looking at the detailed composition of the errors in Figure 8a,b, we observe that the worst outliers are not visible because they make up less than 5% of the data. However, for the first trajectory, an increased error is visible for the $z$-coordinate, which also increases the overall error. This is probably created through the location of the anchor nodes in a plane parallel to the $xy$-plane.

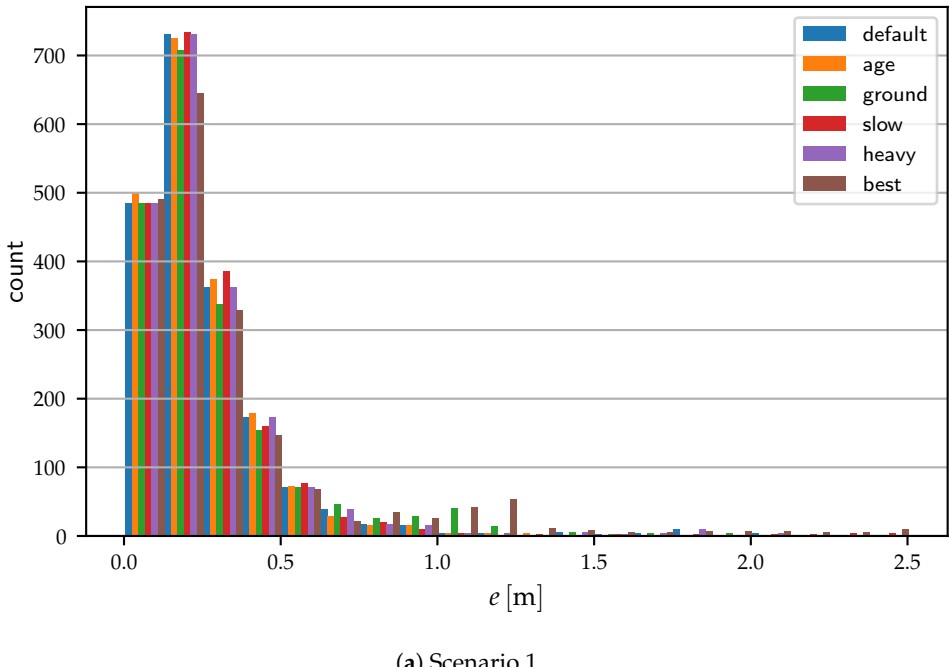

(**a**) Scenario 1.

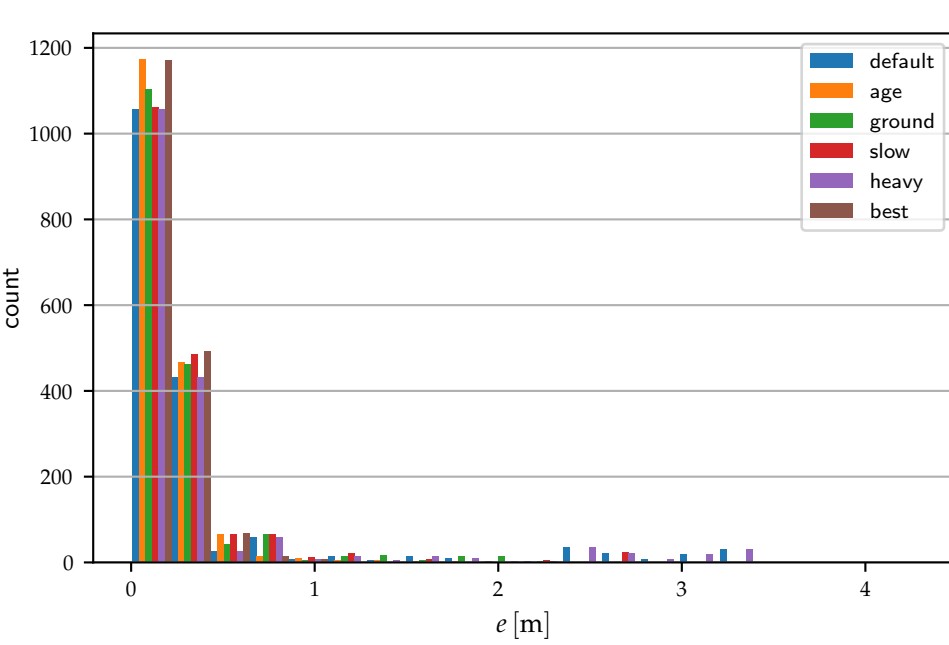

(**b**) Scenario 2.

**Figure 7.** Histogram of localization errors for both indoor evaluation scenarios. Each parameter configuration of the localization algorithm is shown in a separate bar.

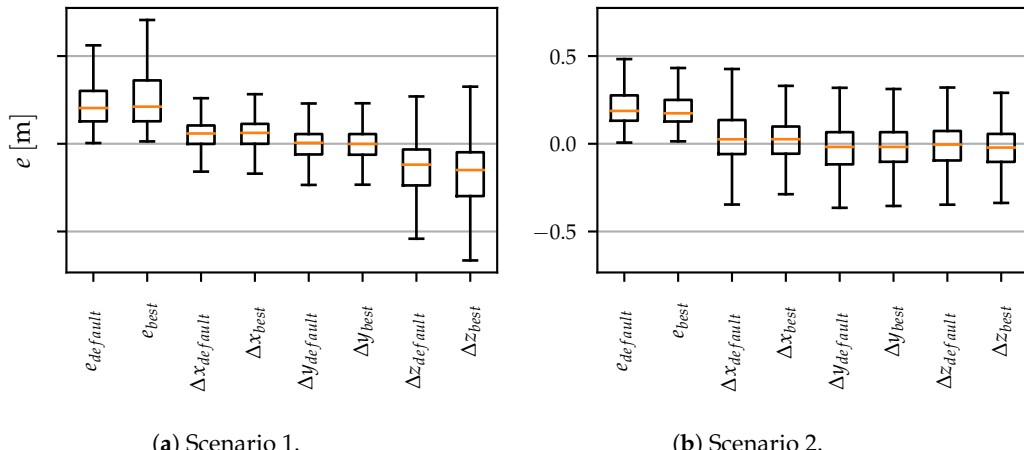

(**a**) Scenario 1.  (**b**) Scenario 2.

**Figure 8.** Box–Whisker plot of the localization error of both indoor scenarios. $e$ shows the norm of the error vector, whereas $x$, $y$ and $z$ show the error for each component of the error vector. The values are shown for the *default* and the *best* parameter configuration of the localization algorithm.

Figure 8a shows that the error in the $z$-coordinate is different from the error in the other components. Based on this situation, we suspected a correlation between the localization error and $z$-coordinate (height). To verify this, we conducted a Pearson correlation analysis between the observed position error $\Delta e$ and the $z$-coordinate of the Vicon reference position. The results are visible in Table 2.

For most algorithm configurations, a correlation coefficient $r_1 \approx -0.4$ for scenario 1 and 2 can be observed. To verify the statistical soundness of the result, we conducted a two-way permutation of 100,000 permutations of the input data for each algorithm configuration for each scenario, which resulted in a $p$-value of 0.0 for all combinations. Consequently, there is a weak negative linear correlation between the position error and the $z$-coordinate of the object to be localized. It appears that the typical ground reflects the wireless signals and generates additional errors through multipath effects. Interestingly, the age configuration using the age-awareness extension of the algorithm reduced the correlation for scenario 2 to $\approx 0.37$. The ground detection with a $z_{min} = 0.0$ m also shows a deduction in the correlation for the second scenario. The slow configuration also reduced the correlation for both scenarios to the minimally observed correlations of $-0.274$ and $-0.25$. However, this comes at the cost of reducing the maximum movement speed of the copters. Interestingly, there is a large deviation in correlation coefficients between scenario 1 and 2 for the best configuration. This is caused by the larger variation of the error in $z$-coordinate for the first scenario with this configuration. In the first scenario, the $z_{min}$ parameter has a large influence and filters out some wrong $z$-coordinates if it is set to 0.2 m. In the second scenario, it does not have any significant impact at all. The best configuration omits the parameter because this makes the resulting algorithm setup more general without any assumptions on anchor placement. Therefore, we decided to use $z_{min} = 0.0$ for best, even though it does not provide the best possible localization precision for both scenarios.

**Table 2.** Pearson correlation coefficients between the error $e$ of localization and $z$-coordinate of the tracked object.

| Configuration | Parameters $r_1$ | $p_1$ | $r_2$ | $p_2$ |
|---|---|---|---|---|
| default | −0.426 | 0.0 | −0.439 | 0.0 |
| age | −0.459 | 0.0 | −0.370 | 0.0 |
| ground | −0.447 | 0.0 | −0.327 | 0.0 |
| slow | −0.274 | 0.0 | −0.250 | 0.0 |
| heavy | −0.426 | 0.0 | −0.439 | 0.0 |
| best | −0.188 | 0.0 | −0.398 | 0.0 |

Figure 9 shows the time difference between received distance measurements of the different anchors. As shown, the time difference is very similar for all nodes. The median is at ≈0.23 s, which indicates that either the receiver module was overloaded with the reception and forwarding of the measurements or wireless packages were lost between the nodes. As the spread of the distribution of time differences is rather high, we assume that we have approx. 50% package loss in our scenario. Consequently, the parameters of the modules need to be optimized to enhance the reliability of the communication.

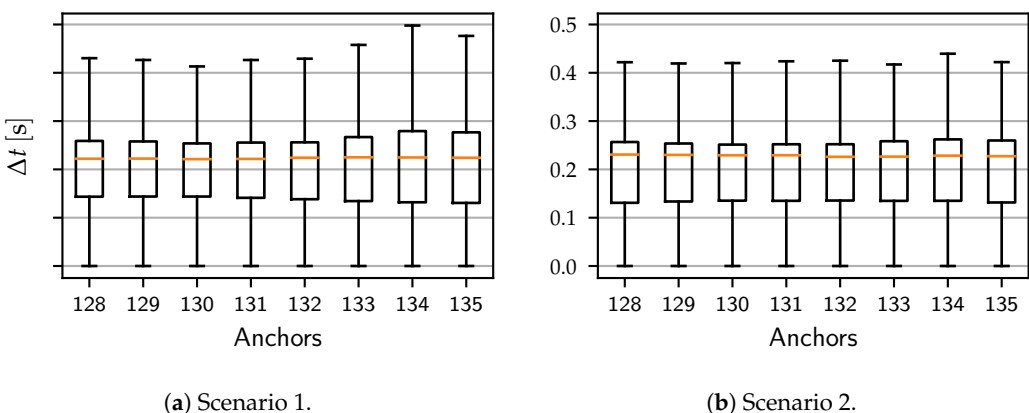

(**a**) Scenario 1.　　　　　　　　　　　　　　　　　　　(**b**) Scenario 2.

**Figure 9.** Box–Whisker plot of the age between received measurements from the different anchors for both indoor scenarios.

The results of our indoor evaluation show that our systems provides similar precision, even though we observed a large loss of packets mid-flight. This, of course, reduces the update rate of the system to less than 10 Hz, which may easily be compensated by the internal INS filters of the autopilot software. Additionally, we achieved an average precision of 0.2 m, which is better than the work of Kempke et al. [18], but with much less communication overhead. This is important because our later extension to a swarm-based localization will increase the necessary communication further. We also achieved slightly less precision than OmniSwarm [20] for two drones. However, this approach used way more sensors and was integrated deeper into the drone system. Our approach can easily be used as an extension module to existing copter builds.

### 5.3. Outdoor Evaluation

In addition to the indoor experiments with a high-precision reference, we conducted two outdoor experiments. These experiments have no high-precision reference. Therefore, we analyzed the behavior based on the flight patterns.

The first experiment used the anchor setup shown in Figure 10. The anchors are all placed in a $xy$-plane because the experiment was conducted on flat ground and no variation in height was possible. The receiving node was attached to a FINken quadcopter, which was manually controlled. The node transmitted all distance measurements to a laptop for logging purposes. The localization algorithm was executed afterwards with configuration best. We executed two trajectories with the same quadcopter. In the first trajectory, the quadcopter hovered over each anchor node with slow movement speed between the anchors. The second trajectory executed multiple counter-clockwise circles over the area.

As shown in Figure 10a, the generic flight behavior can be reconstructed by the localization system. The visible instability is not necessarily caused by the localization system, but may also be caused by the manual control of the copter. The second trajectory, visible in Figure 10b, also shows the expected behavior of counter-clockwise circles in the area.

Similar to the results of the indoor experiments, Figure 11a,b show the higher probability of localization errors close to the ground. In the case of trajectory 1, the $z$-coordinate

of the copter increased to $\approx$2.5 m, even though the flight had not started yet. The final $z$-coordinate after landing (after $\approx$250 s and $\approx$200 s, respectively) also shows an offset of $\approx$1 m. The offset of the $z$-coordinate cannot be evaluated mid-flight because no reliable reference is available for these flights.

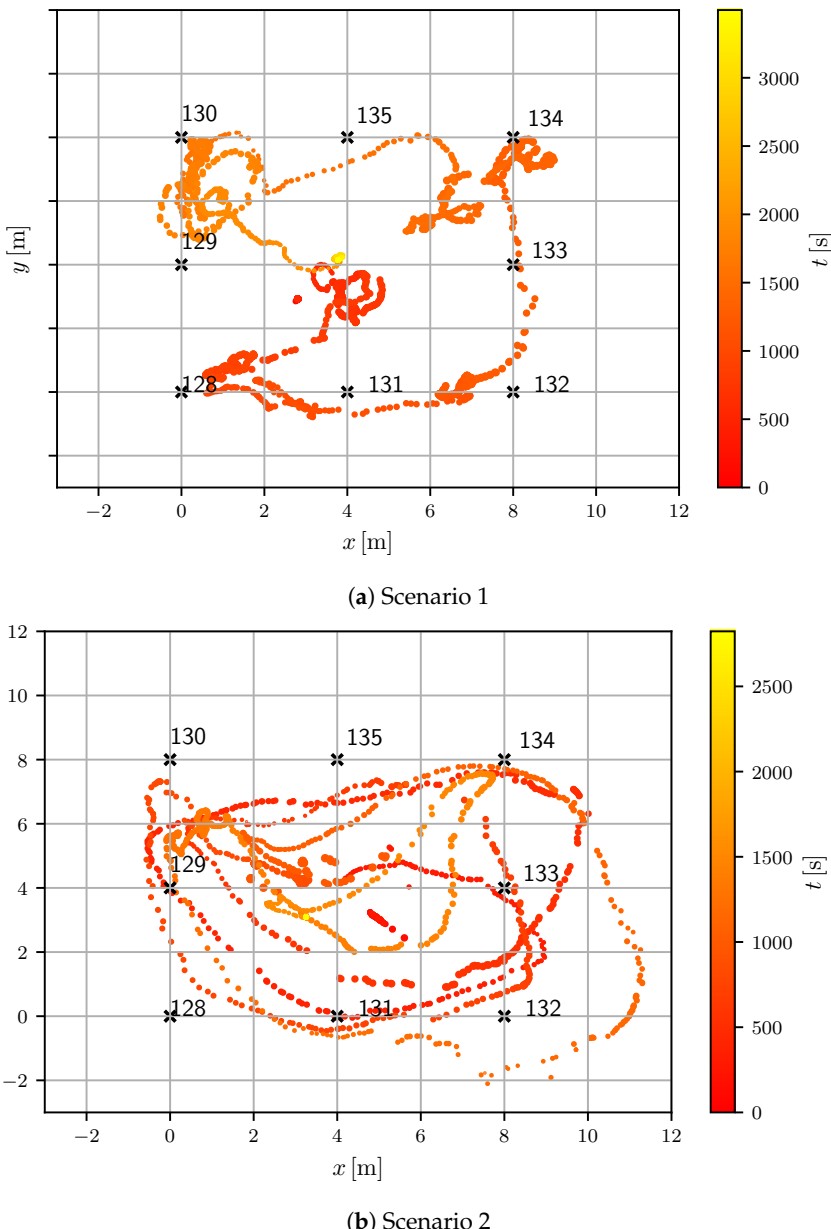

**(a)** Scenario 1

**(b)** Scenario 2

**Figure 10.** Visualizations of the reconstructed trajectories from the first outdoor evaluation for both scenarios. The flight time at each position is shown as a color from red to yellow according to the attached color bar.

As shown in Figure 12, the distribution of the time difference between the reception of distance measurements in both trajectories show similar median values to the indoor experiments. However, the variance between the individual anchor nodes is higher. This may be caused by shadowing effects of the components of the copter, especially the battery, which prevent line-of-sight communication to certain anchors.

The second outdoor experiment used the fully integrated system. The receiver node was attached to a copter. The node transformed the received distance measurements to positions in the local coordinate system using configuration ground without any speed limitation. The resulting relative coordinate was then transformed to GPS coordinates

based on hard-coded reference points. The GPS information was packed into the NMEA format and directly transmitted to the copter via UART. The copter used the provided data, similar to real GPS data. In this experiment, we set up anchors along a walkway in a park close to the faculty. This allowed us to easily obtain the GPS coordinates of the anchors with relatively high precision from open street maps. The resulting anchor position overlaid on top of an OpenStreetMap is shown in Figure 13. The copter was controlled manually and again flew over all anchor nodes, while always hovering above each one.

Unfortunately, the used hard-coded reference points deviated from the anchor positions. Therefore, the references need to be modified after the experiment. To this end, a nonlinear gradient descent on an affine transformation consisting of scaling, rotation and translation was executed. The loss function of the gradient descent was formed by the sum of squared distances between each trajectory position and the position of the associated anchor. The position–anchor association was performed manually and is shown in Figure 14. The resulting trajectory after executing optimized affine transformation on the original trajectory is shown in Figure 13.

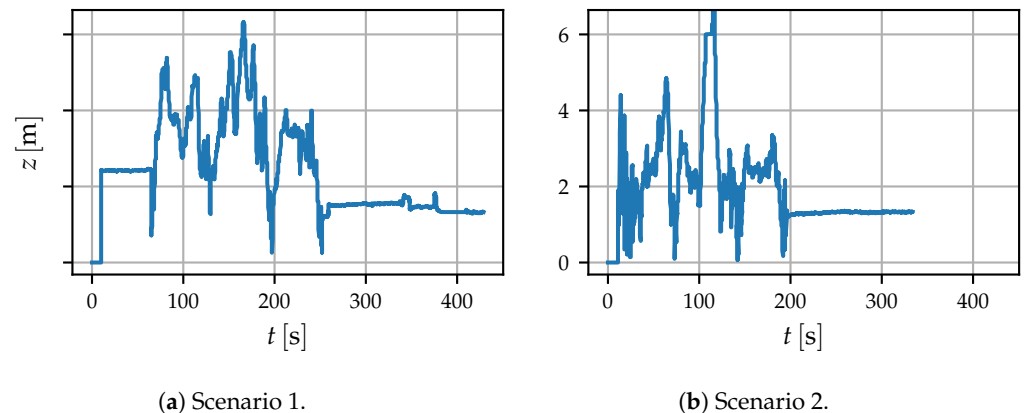

(**a**) Scenario 1.                    (**b**) Scenario 2.

**Figure 11.** Line plot of the reconstructed $z$-coordinates of the first outdoor evaluation experiment for both scenarios.

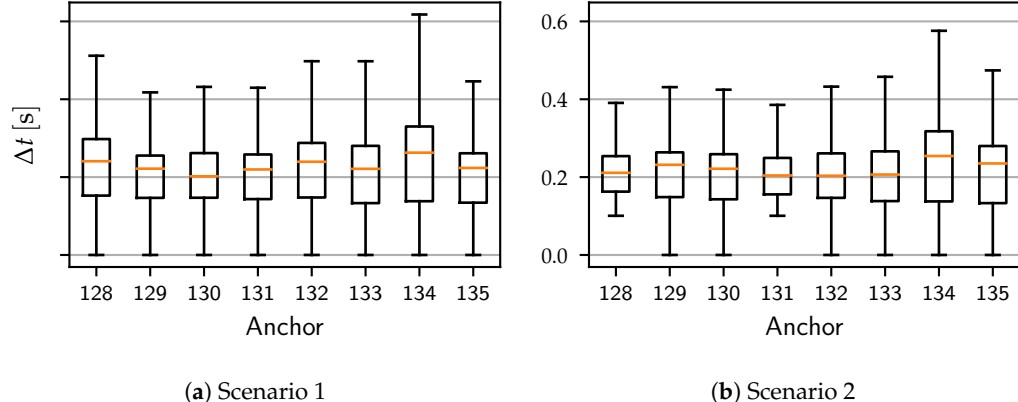

(**a**) Scenario 1                    (**b**) Scenario 2

**Figure 12.** Box–Whisker plot of the distribution of time difference between reception of distance measurements from the different anchors for both scenario of the first outdoor evaluation.

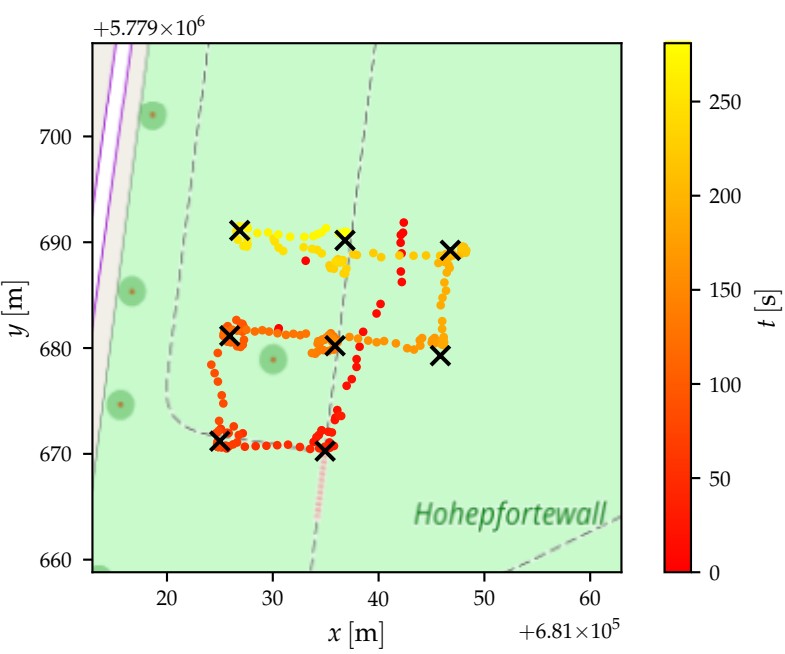

**Figure 13.** Visualization of the anchors, node trajectory and localization error for the second outdoor experiment. The anchors are marked with **X**. The node trajectory is colored to indicate time. The flight time is visible in seconds in the color bar.

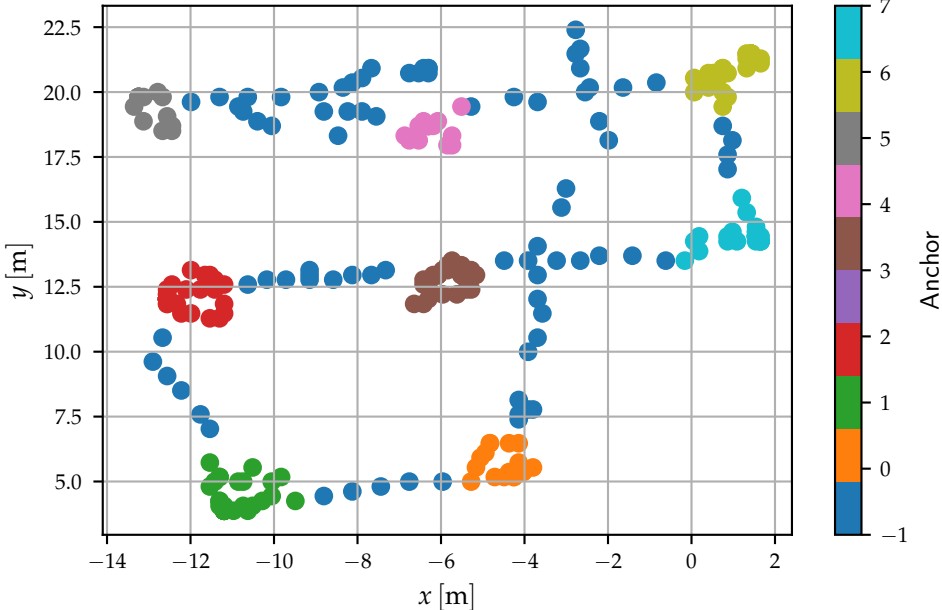

**Figure 14.** Visualization of the anchor assignment for evaluation of the outdoor performance. Each position is assigned to the anchor with the respective color visible in the color bar. The anchor with number −1 is a virtual anchor, capturing all positions not assigned to any anchor.

Figure 15 shows the resulting errors of the associated anchors and the positions. As shown, the general positions of the copter fit the expected trajectory. However, Figures 13 and 15 still show larger deviations from the correct positions. We expect the rotation of the copter together with the antenna characteristics of the node to have a high influence in this scenario.

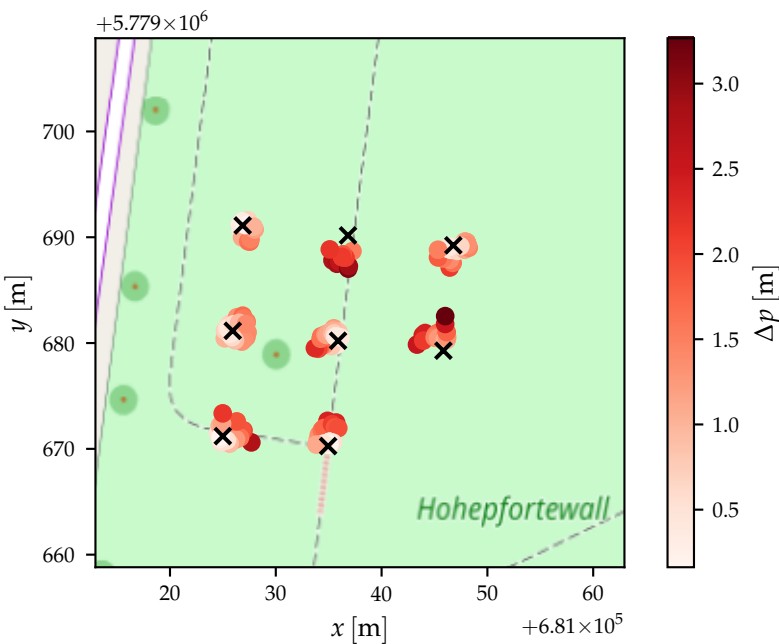

**Figure 15.** Visualization of the anchors and localization error for the second outdoor experiment. The anchors are marked with **X**. The estimated position error of each position is indicated using colors visible in the color bar.

Additionally, the embedded software did not yet support all parameters of the localization algorithm, which is the reason for the usage of a suboptimal configuration.

## 6. Conclusions and Future Work

In this paper, we showed that GPS may be unreliable, and additional local relative localization systems are necessary to support the robust behavior of drones. To this end, we developed a localization system based on UWB-Ranging to fixed anchor nodes. We showed that without additional hardware besides the DWM1000 module, the distance measurements between the quadcopter and the anchors are typically in the range of $\pm 0.2$ m. We developed two algorithms, which together form a continuous localization mechanism usable in autonomous quadcopters. The algorithms were tested with different parameters in indoor and outdoor scenarios. In the indoor scenario, the achievable precision has a median value of $\approx 0.2$ m. The outdoor scenarios showed the system capability regarding the tracking, even though no precise error estimation was possible. An interesting finding of the approach was that the error in the localization has a weak correlation to the height of the object being tracked. This seems to be created by reflection and line-of-sight obstruction created by the ground.

In summary, the described approach showed reasonable performance on par or better than state-of-the-art approaches, but with minimal integration to the drone system. This enables easy integration to other copters as long as they are using the same autopilot software. Additionally, the approach is very efficient regarding communications, which is beneficial for the later extension to a swarm-localization system.

For future work, we want to integrate information from the INS of the copters. This enables the additional usage of movement commands and local sensors on the copter to further increase the precision of the localization. While the used NMEA protocol is rather simple, it is limited in update rate and precision. However, an implementation of the UBLOX protocol is planned. This will provide a higher precision and update rate. In the next step, the embedded localization software will be enhanced to support the *best* parameter configuration to enhance localization accuracy. The third modification that we

aim to evaluate is the usage of external antenna for the DWM100 modules to mitigate the line-of-sight issues in close-to-ground scenarios.

Our final goal is the extension of the current system to multihop localization to enable swarm localization of a swarm of drones. To this end, the localization node of each swarm member needs to be able to freely switch between the tracked node and anchor node based on its possible contribution to the localization quality of other localization nodes. The benefit of this approach is that even a copter with depleted energy, which cannot continue flying, may serve the swarm as an additional anchor for localization. To this end, a decision-making algorithm needs to be developed, which can manage the different goals of such an integrated swarm localization system, such as minimizing movement times, maximizing localization precision, ensuring safety of each drone and minimizing energy cost.

**Author Contributions:** Conceptualization, C.S.; methodology, C.S. and J.B.; software, C.S. and J.B.; validation, C.S. and J.B.; formal analysis, C.S.; resources, S.M.; data curation, J.B.; writing—original draft preparation, C.S. and J.B.; writing—review and editing, C.S. and S.M.; visualization, C.S.; supervision, S.M.; project administration, S.M. All authors have read and agreed to the published version of the manuscript.

**Funding:** This research received no external funding.

**Institutional Review Board Statement:** Not applicable.

**Informed Consent Statement:** Not applicable.

**Data Availability Statement:** The raw data generated in the experimental evaluation are available in the GitHub repository `ovgu-FINken/DWM1000_Copter_Integration` in the branch `drones21` starting from commit `ab2a713f28c8440dbf8cdc8bbb45144af2c7ad26` in folder `data`.

**Conflicts of Interest:** The authors declare no conflicts of interest.

## Abbreviations

The following abbreviations are used in this manuscript.

| | |
|---|---|
| UWB | Ultrawideband |
| GPS | Global Positioning System |
| AGPS | Assisted GPS |
| D-GPS | Differential GPS |
| ToF | Time-of-Flight |
| TDoA | Time-Difference of Arrival |
| RSSI | Received Signal Strength |
| SD-Card | Secure Digital Card |
| UAV | Unmanned Aerial Vehicle |
| USART | Universal Serial Asynchronous Receiver Transmitter |
| I$^2$C | Inter-Integrated Circuit |
| NMEA | National Marine Electronics Association |
| UTM | Universal Transverse Mercator |

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
