# Peer review of "A Single-Copter UWB-Ranging-Based Localization System Extendable to a Swarm of Drones"

_drones, doi:10.3390/drones5030085_

Round 1

Reviewer 1 Report

The proposed paper is devoted to a topical actively developing research area of solving the problem of wireless drone localization.

Despite the fact that the work is called ‘Towards an UWB-Ranging-based Localization for Swarms of Drones’ no specificity related to the swarm behavior of drones is being studied in the work.

The work analyzes the positioning accuracy of a single drone with DWM1000 module used for positioning. However, a lot of works (some of which the authors refer to) are devoted to the use of this module for drones positioning and approximately the same characteristics are obtained. In the paper the results obtained does not compare with the results obtained in this area by other research teams.

It is not clear from the text of the work which new quality the authors managed to achieve when using this module, which fundamentally distinguishes this work from others.

What are the features of this technical solution, which can give advantages when moving to the swarm configuration?

The paper represents some groundwork for future work, which is undoubtedly useful for a particular scientific group, but the independent value of the presented results is questionable.

There are also a number of comments on the text and design of the paper:

  1. Figure 2 (a): the figure does not provide any useful information. It is enough to indicate the dimensions of the board in the text of the article or in fig.2 (b).
  2. Line 121 “… 3.5 Hz to 6.5 Hz …”: should be “3.5 GHz to 6.5 GHz”.
  3. The paper text says that the anchors were placed at a height of 0.2 m. And what does zmin = 0 mean for the ‘ground’ and ‘best’ scenario? The anchor was placed on the ground at a height of 0 m? Or is this value only used in a numerical method? Then what explains the obtained advantage? In the text, it was argued that one should take zmin = 0.2? The article does not declare this situation in any way.
  4. Figure 5 (b): Must be “Scenario 2” instead of “Scenario 1”
  5. Line 295 “Figure 7 shows that the errors in z-coordinate is different from error in the other components.”: There should be Figure 7 (a), because in fig. 7 (b) there is no difference between the z coordinate and other coordinates. And by the way, it is not clear why in this case the correlation r2 even greater than r1 (for configuration best).
  6. Lines 341-342: It is not clear whether offset persists during flight or occurs only at the surface.
  7. Figure 13: Anchor markers ‘X’ mentioned in figure caption are absent in the figure.
  8. The design of the paper is unsuccessful. The sequence of the appearance of the pictures is violated. Reference to fig. 8 appears in the text between the references to fig. 4 and fig. 5. In this case, the drawing itself appears after 3 pages. Tables 1, 2 fig. 5,6,7 appear before they are referenced in the text. This all confuses and complicates the perception of the paper. Reference to fig. 12 is absent. Reference to fig.13 is after the reference to fig. 14.

It is recommended to revise the article and make the presentation of the scientific novelty and significance of the results obtained more transparent.

Reviewer 2 Report

This paper cannot be accepted in its current format, as it introduces the work as a swarm-based localization in the title and introduction, but then it is only presenting a single drone localization system, both in the algorithmic part and in the experimental evaluation.

Reviewer 3 Report

Dear authors. I am recommending your paper to be accepted. I would like to motivate my opinion and offer some suggestions.

Abstract: fine
Section 1: Short and concise but to the point, well done.
Section 2: 4 approaches are discussed, again, short and concise, well done.
Section 3: I am not an expert in the technical systems but this also seems adequate and well written.
Section 4: All there, well done.
Section 5: Experiments and results are discussed. Some re-layouting and shuffeling of Figures might improve readability, adding captions may further improve accessibility and understanding of the paper.
Conclusion: To the point

My high level comment for the paper is this: you showed that GPS is "sometimes unreliable" which is known. You provided the outcomes of your investigations to show this unreliability (a) as confirmation of the 'known' issue as well as (b) to set the scene for the testing of your approaches. You have then provided two algorithms to overcome the issue, specifically for quad-copters (or any copter I guess) and tested the approaches in an inside and an outside setting. The results look good and I understand the discussion thereof. You yourself list a number of next steps and future work ideas, indicating that this work is in fact part of a 'plan' and that the results you generated are, in fact, needed for your work. I like that this places the work in a greater (and believable) context. I consider the paper acceptable and will recommend it to be accepted.

However, your paper specifically claims to be about swarms and I am missing this aspect. You do state that you want to use this for a swarm, and you motivate the need for localization in a swarm. But you only flew a single device (if I understand the paper correctly). As far as I understand, the concept of the swarm only comes into play when all devices but the one we are interested in are serving as anchor nodes. I would like to request that you either (a) add a page or so to detail the connection to the swarm or (b) change the title and the introduction. Of course, you can (c) respond by pointing out where I missed / misread the relevant paragraph :-). I am primarily concerned about managing expectations for the reader. I read the paper with different expectations and assumptions and only when going through it in detail did I understand how to interpret your use of "swarm" in the title.

General comment: you often use "we", this might be considered too informal by some. This is just a general comment, feel free to ignore this.

In the abstract, you write that "additional copter and ... are done". Do you mean "additional copters are added"?

In line 51 you write that the "field is very fast" ... I assume you mean "progressing fast" or "advancing fast"?

On page 4, Figure 2: can you rotate the right image and make it larger? It seems that the left image does not have fine details that need to be shown that large, while the right image could be a bit larger.

In line 200, it should be Algorithm 1 (as opposed to Section 1), right? Or are you indeed referring to the introduction? If I am correct then you need to rethink the wording because "Algorithm 2 describes the process [described in Algorithm 1]" sounds off. Short: please check this and make sure this is saying what you want it to say.

In line 212 you reference a line from Alg 2. You use a different font to do so, but this is not matched by the font used in the Algorithm itself. Make this consistent for improved readability.

In line 243 you state "most of the time". To make this a bit less ambiguous, is it correct to add something along the lines of "but always at least 4"? The image in Figure 3 seems to suggest that the minimum is 4.

In line 245 you could replace the word inhomogeneous with heterogeneous ... (use your own judgement)

In line 251 you state that there are benefits wrt "swarm stability". This concept, while referenced, is not explained in the paper. Maybe you could provide a brief description of that this means (it could be added in the introduction where you reference the literature).

For table 1 you could (use your own judgement) indicate the columns where values do change. Out of the 9 columns, 3 do not change and 2 only change (by one order of magnitude) once: for the config heavy. This is just a comment, feel free to ignore but it might help readability. In line 267 you mention nu (which is consistently set to 0.1) as well as p(0) (line 268) which is also the same. You could just mention this in the text and reduce the data provided in the table.
Furthermore, I personally like the captions of a table / figure to provide enough context to allow for an informed first glance at the table.

Reviewer 4 Report

This work indicates a drone localization system as the first step towards a scalable multi-hop drone swarm. The drones in this study have applied Ultra-Wideband transceiver modules for communication and distance measurement. The positions of the drones have been identified based on fixed anchor points. The distance between each drone and the anchors has been measured periodically, and a tailored gradient descend algorithm has been applied to solve the resulting non-linear optimization problem. Moreover, the location of each drone has been tracked in every time step. The paper is well-written, has high quality and can be accepted for publication in the “Drone” in its present form.

Round 2

Reviewer 1 Report

Remarks taken into account. Article can be accepted.
